# Nanoconfinement facilitates reactions of carbon dioxide in supercritical water

Nore Stolte [1,4], Rui Hou [1,2] & Ding Pan [1,2,3] ✉

The reactions of $CO_2$ in water under extreme pressure-temperature conditions are of great importance to the carbon storage and transport below Earth's surface, which substantially affect the carbon budget in the atmosphere. Previous studies focus on the $CO_2$(aq) solutions in the bulk phase, but underground aqueous solutions are often confined to the nanoscale, and nanoconfinement and solid-liquid interfaces may substantially affect chemical speciation and reaction mechanisms, which are poorly known on the molecular scale. Here, we apply extensive ab initio molecular dynamics simulations to study aqueous carbon solutions nanoconfined by graphene and stishovite ($SiO_2$) at 10 GPa and 1000 ~ 1400 K. We find that $CO_2$(aq) reacts more in nanoconfinement than in bulk. The stishovite-water interface makes the solutions more acidic, which shifts the chemical equilibria, and the interface chemistry also significantly affects the reaction mechanisms. Our findings suggest that $CO_2$(aq) in deep Earth is more active than previously thought, and confining $CO_2$ and water in nanopores may enhance the efficiency of mineral carbonation.

Aqueous fluids play a critical role in transporting carbon between Earth's surface and interior[1–3], which is a substantial part of Earth's carbon cycle, with great implications for global climate and human energy consumption. It has long been assumed that aqueous carbon solutions under extreme pressure (P) and temperature (T) conditions are made by mixtures of neutral gas molecules[4], e.g., $H_2O$, $CO_2$, $CH_4$; however, recent studies showed that important chemical reactions occur between water and carbon species, resulting in significant amounts of ionic products, which may further participate in water-rock interactions and the formation of diamonds in Earth's interior[5–11]. Most of the previous studies focus on the properties of aqueous carbon solutions in the bulk phase. In fact, aqueous solutions in deep Earth are often confined to the nanoscale in pores, grain boundaries, and fractures of Earth's materials[12–14], where the physical and chemical properties of solutions may be dramatically different from those of bulk solutions. In addition, in carbon capture and sequestration efforts, $CO_2$ mineralization occurring in water trapped in porous rocks offers an efficient and secure method to permanently store carbon underground with a low risk of return to the atmosphere[15]. The behavior of aqueous carbon solutions under nanoconfinement at extreme P-T conditions is of great importance to the deep carbon cycle and $CO_2$ storage, but is poorly understood on the molecular scale.

Previous studies reported that nanoconfinement substantially affects properties of water, e.g., equation of state[16–18], phase behavior[19–21], dielectric constant[22–26], and diffusion[27–29]; as a result, the reactivity of solutes under confinement may be very different from that in bulk solutions[30]. The dimensional reduction and increased fluid density could enhance reactions between small solutes in nanoconfinement[31,32], whereas reactions involving large reactants or intermediates may be sterically hindered[33]. Further, the increase of the dielectric constant of nanoconfined water parallel to the confining surface leads to the stabilization of aqueous reaction products with charges[33], causing the enhanced autodissociation of water[23]. The solid−liquid interface also greatly affects the properties of confined aqueous solutions[34]. Preferential adsorption of solutes at the confining interface may shift reaction equilibria. For example, in the production

[1]Department of Physics, Hong Kong University of Science and Technology, Hong Kong, China. [2]HKUST Shenzhen-Hong Kong Collaborative Innovation Research Institute, Shenzhen, China. [3]Department of Chemistry, Hong Kong University of Science and Technology, Hong Kong, China. [4]Present address: Lehrstuhl für Theoretische Chemie, Ruhr-Universität Bochum, 44780 Bochum, Germany. ✉e-mail: dingpan@ust.hk

of methane from carbon dioxide at hydrothermal vent conditions ($CO_2 + 4 H_2 \rightleftharpoons CH_4 + 2 H_2O$), hydrophilic pore surfaces adsorb water, favoring the production of methane[35].

Nanoconfinement and interface chemistry may both likely change the properties of aqueous carbon solutions, but a molecular understanding is lacking on how chemical speciation and reaction mechanisms are affected. It was experimentally found that magnesite precipitates much faster in nanoscale water films than in bulk water[36]. Because it is very challenging to study aqueous solutions under nanoconfinement in experiment, atomistic simulations are widely used. Many studies applied classical force fields[27,29,34,37], which were usually designed for bulk solutions at ambient conditions; their accuracy at extreme conditions is not well tested. As a comparison, ab initio molecular dynamics (AIMD) simulations do not rely on experimental input or empirical parameters[38–40]. We solve the many-body electronic structure numerically, so the breaking and forming of chemical bonds, electronic polarizability, and charge transfer are all treated at the quantum mechanical level[40,41]. The AIMD method is widely considered as one of the most reliable methods to make predictions, and many simulation results were later confirmed by experiments[40,41].

Here, we performed extensively long AIMD simulations to study $CO_2(aq)$ solutions nanoconfined by graphene and stishovite ($SiO_2$) at 10 GPa and 1000 - 1400 K. These P-T conditions are typically found in Earth's upper mantle. We compared the $CO_2(aq)$ reactions in nanoconfinement with those in the bulk solutions, and examined how weak and strong interactions between confining walls and confined solutions affect chemical speciation and reaction mechanisms. Although graphene is not found in deep Earth so far, it provides a good comparison with stishovite. In graphene confinement, there are no chemical reactions between graphene and solutions, whereas the dangling atoms in stishovite actively participate in aqueous carbon reactions, so we can compare the effects of spatial confinement with and without interface chemistry. What's more, thanks to the rapid development in the fabrication and characterization of 2D materials in recent years, experimentalists are now able to delicately measure the properties of aqueous solutions under graphene nanoconfinement[30], so we hope our study can also attract many follow-up experiments. Our work is relevant to the carbon transformation in deep Earth, and also helps us to understand atomistic mechanisms of $CO_2$ mineralization in the carbon capture and storage.

## Results and discussion

### Graphene nanoconfinement

We first studied $CO_2(aq)$ solutions confined by two graphene sheets at ~10 GPa, and 1000 ~ 1400 K (Fig. 1a). The graphene sheet separation was 9.0 and 9.2 Å at 1000 and 1400 K, respectively. We modeled the graphene sheets using a distance-dependent potential acting on the carbon and oxygen atoms, which was fitted to the interaction energies calculated using diffusion quantum Monte Carlo[42] and van der Waals density functional theory[43] (see Supplementary Methods). We calculated the pressure of confined solutions parallel to the graphene sheets, which is ~10 GPa (see Supplementary Methods). In addition, we also used atom number density profiles to calculate actual volumes that aqueous carbon solutions occupy, and then applied the equation of state of $CO_2$ and water mixtures to obtain the pressure[44].

We directly dissolved $CO_2$ molecules in the supercritical water, and the initial mole fraction of $CO_2(aq)$ is 0.185. The $CO_2$ molecules reacted frequently with water, and we performed long AIMD simulations until the concentrations of carbon species reached equilibria (see Supplementary Fig. 3). Initially, the reaction between $CO_2(aq)$ and $H_2O$ produces bicarbonate ions ($HCO_3^-$):

$$CO_2(aq) + 2H_2O \rightleftharpoons HCO_3^- + H_3O^+. \tag{1}$$

This reaction in some cases occurs in one step, or involves the dissociation of water so that $OH^-$ can react with $CO_2(aq)$ to form $HCO_3^-(aq)$. The generated bicarbonate ion may further accept a proton to become a carbonic acid molecule ($H_2CO_3(aq)$), or may lose a proton to become a carbonate ion ($CO_3^{2-}(aq)$). The major carbon species in the solutions are $CO_2(aq)$, $CO_3^{2-}$, $HCO_3^-$, and $H_2CO_3$.

We compared the chemical speciation of the solutions under nanoconfinement and in the bulk phase at the same P-T conditions[10]. Figure 2 shows that at 1000 K, the mole percent of $CO_2(aq)$ in total dissolved carbon under nanoconfinement is $1.3 \pm 0.9\%$, whereas it is $15.2 \pm 2.0\%$ in the bulk solution. The mole percent of $HCO_3^-$ under nanoconfinement ($50.0 \pm 1.0\%$) is higher than that in the bulk solution ($35.9 \pm 0.7\%$), and the concentrations of $H_2CO_3(aq)$ are similar ($42.7 \pm 1.7\%$ vs. $46.8 \pm 1.5\%$). With increasing temperature from 1000 K to 1400 K, the mole percents of $CO_2(aq)$ under nanoconfinement and in the bulk solution increase to $14.5 \pm 3.2\%$ and $58.8 \pm 2.0\%$, respectively. The equilibrium concentrations of $CO_2(aq)$ in the nanoconfined

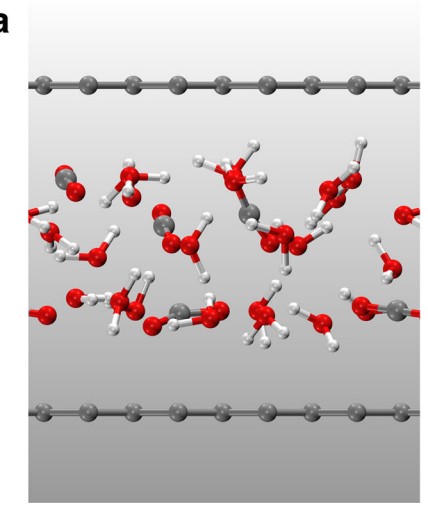

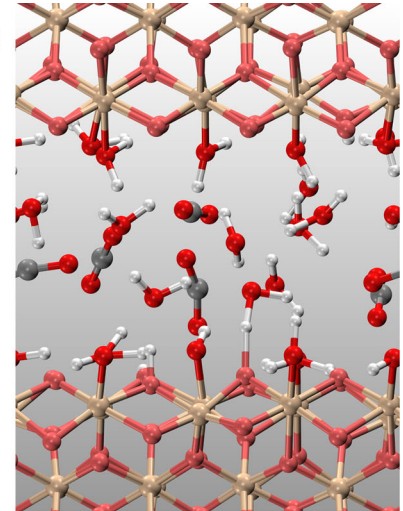

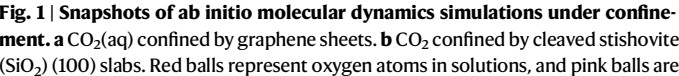

**Fig. 1 | Snapshots of ab initio molecular dynamics simulations under confinement. a** $CO_2(aq)$ confined by graphene sheets. **b** $CO_2$ confined by cleaved stishovite ($SiO_2$) (100) slabs. Red balls represent oxygen atoms in solutions, and pink balls are oxygen atoms in $SiO_2$. Gray, white, and yellow balls are carbon, hydrogen, and silicon atoms, respectively.

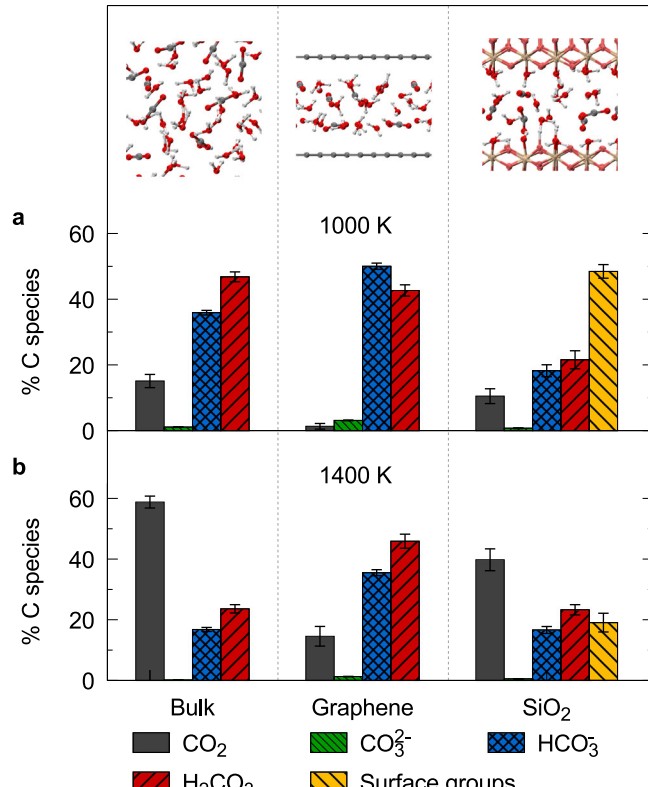

**Fig. 2 | Mole percents of carbon species in the CO₂(aq) solutions in bulk and nanoconfined by graphene and stishovite (SiO₂) at chemical equilibria.** The initial mole fraction of $CO_2$(aq) is 0.185. The pressure is -10 GPa. The temperatures are **a** 1000 K and **b** 1400 K. The data of bulk solutions in (**a**) are from ref. 10, and the bulk data in (**b**) were interpolated using the simulation results in ref. 10. Uncertainties were obtained using the blocking method[68].

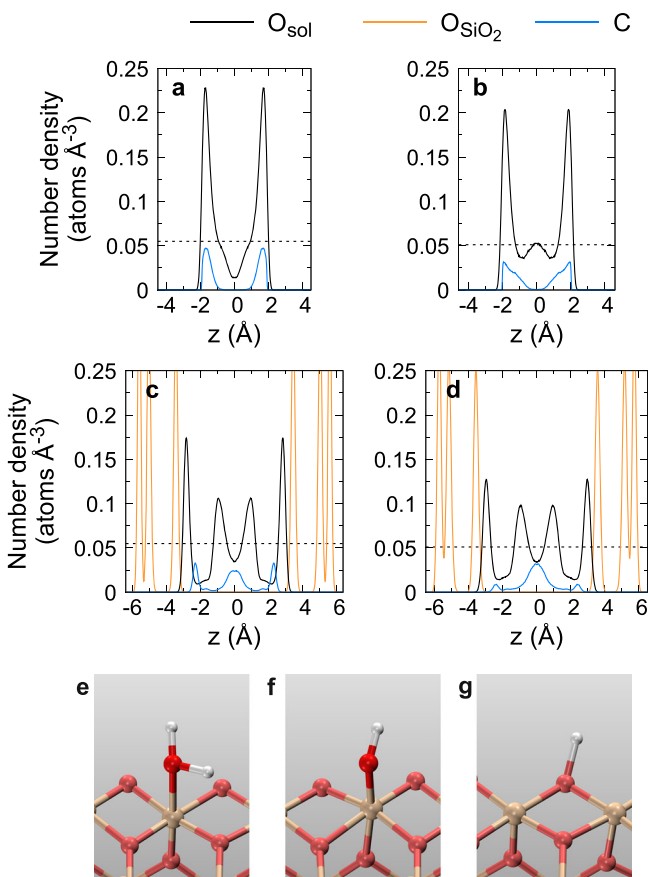

**Fig. 3 | Number density profiles of oxygen atoms and carbon atoms along the *z* axis, normal to the confining surfaces. a**, **b** The solutions under graphene confinement at 1000 and 1400 K, respectively. **c**, **d** The solutions under stishovite confinement at 1000 and 1400 K, respectively. $O_{sol}$ refers to the oxygen atoms in solutions (black lines), $O_{SiO_2}$ refers to the oxygen atoms in stishovite (orange lines), and C refers to carbon atoms (blue lines). The initial mole fraction of $CO_2$(aq) is 0.185. The pressure is -10 GPa. The center of confined fluids is set at $z = 0$, and the density distributions have been symmetrized. The horizontal black dashed lines represent the oxygen density in the bulk solutions at the corresponding P-T conditions[44]. **e**–**g** The $H_2O$ molecule, the $OH^-$, and $H^+$ ions bonded to the stishovite (100) surface, respectively.

solutions are lower than those in the bulk solutions at the two temperature conditions studied here, suggesting that nanoconfinement promotes the $CO_2$(aq) reactions. When increasing temperature along an isobar, due to thermal entropy effects, small molecules like $CO_2$(aq) are more favored. We did not see obvious difference in reaction rates between nanoconfined and bulk solutions.

To understand why nanoconfinement promotes $CO_2$(aq) reactions, we analyzed the water structure in Fig. 3. In the graphene-confined solutions, there are two sharp density peaks for oxygen atoms, corresponding to two water layers (Fig. 3a, b). We found that the carbon-containing ions and molecules in these two layers tend to align parallel to the graphene sheets (Fig. 4), and $CO_2$(aq) mostly reacts with water molecules in the same layer (Supplementary Figs. 6 and 7). Nanoconfinement increases the probability of reactive encounters between $CO_2$(aq) and solvent molecules, as diffusion is restricted to two dimensions[33]. It has been reported that the dielectric constant of nanoconfined water in the direction parallel to the confining surfaces ($\epsilon_\parallel$) increases significantly compared to the bulk value ($\epsilon_0$). In water, the Coulomb interaction between two ions is $F = \frac{q_1 q_2}{\epsilon_0 r}$, where $q_1$ and $q_2$ are the charges of the two ions, and $r$ is their distance. With increasing the dielectric constant, the magnitude of $F$ decreases, so it is easier to separate a cation from an anion. Consistent with this, water molecules dissociate more easily under nanoconfinement[23,25]. In $CO_2$(aq) solutions, the produced $OH^-$ ions from the water self-ionization are subsequently available to react with $CO_2$(aq) (reaction (1)). The enhancement of $\epsilon_\parallel$ also further stabilizes $HCO_3^-$ and $CO_3^{2-}$ ions generated in the reaction between $CO_2$(aq) and $H_2O$ or $OH^-$. As a result, more $CO_2$(aq) molecules react under nanoconfinement than in bulk. When the interlayer distance between graphene sheets increases

beyond -1.5 nm, the bulk behavior of water is recovered in the center of the slit pore and the effects of nanoconfinement become less obvious[25].

## Stishovite nanoconfinement

After studying the effects of graphene nanoconfinement, we turned to the confinement by a realistic mineral in deep Earth, stishovite, which is a stable phase of $SiO_2$ (space group: P4₂/mnm) at the P-T conditions studied here[45,46] and a major component of subducted oceanic crust[47], playing a substantial role in transporting water into Earth's mantle[48]. We exposed the cleaved stishovite (100) face, one of the low-energy surfaces[49], to the carbon solutions as shown in Fig. 1b. We carried out constant-pressure (NPT) simulations to keep the pressure perpendicular to the solid–liquid interface at -10 GPa, and then we found that the distance between the outermost oxygen atoms in two stishovite (100) surfaces is -7 Å (Supplementary Table 1). In deep Earth, aqueous solutions released from subducting materials in devolatilization processes tend to locate at grain boundaries[14]. The size of the pores at grain boundaries is typically between 0.4 and 1.2 nm[13], and our confinement width is within this range.

Figure 2 shows the chemical speciation of aqueous carbon solutions under stishovite confinement at -10 GPa and 1000 - 1400 K. We

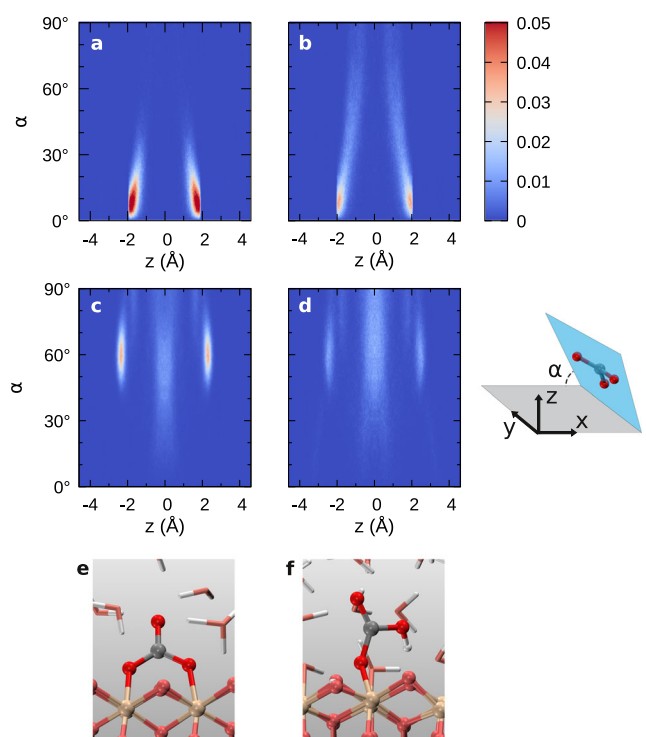

**Fig. 4 | Orientation distribution of sp² carbon species in CO₂(aq) solutions.** The dihedral angle $\alpha$ is between the confinement interface and the plane defined by the three oxygen atoms in sp² carbon species. **a, b** The solutions under graphene confinement at 1000 and 1400 K, respectively. **c, d** The solutions under stishovite confinement at 1000 and 1400 K, respectively. The pressure in all solutions are ~10 GPa. The initial mole fraction of CO₂(aq) is 0.185. The center of confined fluids is set at $z = 0$, and the angle distributions have been symmetrized. **e, f** The $CO_3^{2-}$ and $HCO_3^-$ ions adsorbed on the stishovite (100) surface, respectively.

found that at chemical equilibrium, $48.5 \pm 2.1\%$ (1000 K) and $19.1 \pm 3.1\%$ (1400 K) of carbon species, mostly $HCO_3^-$ and $CO_3^{2-}$, are bonded to the stishovite surfaces, unlike in the graphene-confined solutions. In the atomic density profiles shown in Fig. 3c, d, there are oxygen and carbon density peaks near the $SiO_2$ surfaces, where the oxygen and carbon atoms come from the solutions, indicating that the solid–liquid interface plays an important role.

In bulk stishovite crystals, silicon atoms are octahedrally coordinated, and oxygen atoms are trigonally coordinated, whereas at the cleaved stishovite (100) surface, silicon atoms form bonds with five oxygen atoms, and each oxygen atom bridges two undercoordinated silicon atoms. Water molecules can directly bond to the stishovite surface, or dissociate under the influence of the surface. The hydroxide ion ($OH^-$) from water dissociation can bond to an undercoordinated silicon atom to form a silanol (Si-OH) group, and the extra proton can bond with the surface oxygen atom to become a Si-(OH⁺)-Si bridge (Fig. 3e–g). Similar hydroxylation occurs at the quartz (1000) surface[50,51]. In our simulations, we found reactions between hydroxyl groups at the $SiO_2$ surface and CO₂(aq) in the solutions forming $HCO_3^-$. Figure 5a shows the reaction snapshots at 1000 K. The surface hydroxyl groups or the undercoordinated oxygen atoms also accept protons released in reaction (1), driving the reaction forward (Fig. 5b).

We analyzed the spatial orientation of the sp² carbon species such as $CO_3^{2-}$(aq), $HCO_3^-$(aq), and $H_2CO_3$(aq) in the stishovite-confined solutions in Fig. 4c, d. We found that the molecular plane of the sp² carbon species bonded to the stishovite surface tends to form an angle of ~60° or ~90° with the solid–liquid interface plane, dramatically different from the orientation of carbon species in the graphene-confined solutions. When the angle is ~60°, the carbon species has two Si–O bonds by straddling two silicone atoms (Fig. 4e), while with the angle

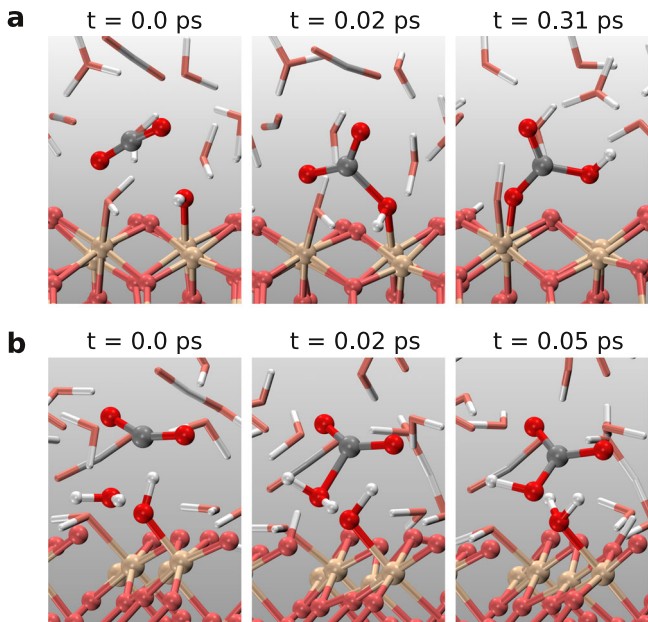

**Fig. 5 | Reactions of CO₂(aq) catalyzed by the stishovite (100) surface. a** The formation of $HCO_3^-$ at the interface. **b** The proton released from the reaction between CO₂(aq) and water is accepted by the silanol (Si-OH) surface group.

of ~90° the carbon species forms only one Si–O bond (Fig. 4f). The strong solid–liquid interaction substantially affects the molecular structure of the confined carbon solutions.

After analyzing the carbon species at the solid–liquid interface, we investigated the carbon species not bonded to the stishovite surface, i.e., fully dissolved in the stishovite-confined solutions. Figure 2 shows that at 1000 K, the mole percent of dissolved CO₂(aq) is $10.5 \pm 2.3\%$, which is larger than $1.3 \pm 0.9\%$ in the graphene-confined solution, and slightly smaller than $15.1 \pm 2.0\%$ in the bulk solution. At 1400 K, the mole percent of dissolved CO₂(aq) in the stishovite-confined solution ($39.8 \pm 3.6\%$) is also between those in the graphene-confined ($14.5 \pm 3.2\%$) and bulk ($58.8 \pm 2.0\%$) solutions.

Both hydroxide ions and protons can be chemically adsorbed on the $SiO_2$ surface, which affects the acidity of carbon solutions. Considering that the pH value of neutral water is no longer 7 under extreme P-T conditions, we calculated the difference between pH and pOH to quantify the acidity of solutions[52]:

$$f = pH - pOH = -\log_{10}\left(\frac{[H_3O^+]}{[OH^-]}\right), \quad (2)$$

where $[H_3O^+]$ and $[OH^-]$ are the concentrations of hydronium and hydroxide ions, respectively. Because CO₂(aq) reacts with water to generate $H_3O^+$, the solutions studied here are all acidic, i.e., $f < 0$, as shown in Table 1. The interesting finding is that the $f$ value of stishovite-confined solutions is more negative than that of graphene-confined solutions at the same P-T conditions, which means that the former is more acidic than the latter, even though less CO₂(aq) reacts in the stishovite-confined solutions. We have discussed that the stishovite surface adsorbs the hydroxide ions and protons from solutions. In addition, our AIMD trajectories show that the $SiO_2$ surface favors the adsorption of $OH^-$ over that of $H^+$ (see Supplementary Fig. 8); as a result, the stishovite-confined solutions are more acidic than the graphene-confined ones at the same P-T conditions. Increasing the concentration of $H_3O^+$ shifts the equilibrium of reaction (1) towards the left, so there is more CO₂(aq) in the solutions.

The nanoconfinement enhances $\epsilon_\parallel$, which stabilized charged ions, so in both graphene- and stishovite-confined solutions, more CO₂(aq)

**Table 1 | Acidity of aqueous carbon solutions: $f$ = pH - pOH**

| T | Confinement | $f$ |
|---|---|---|
| 1000 K | Graphene | $-1.24 \pm 0.02$ |
| | Stishovite | $-1.46 \pm 0.02$ |
| 1400 K | Graphene | $-0.94 \pm 0.05$ |
| | Stishovite | $-1.13 \pm 0.03$ |

The initial mole fraction of $CO_2$(aq) is 0.185, and the pressure is ~10 GPa. Uncertainties are obtained using the blocking method[68].

reacts than in the bulk solutions. However, it has been reported that $\epsilon_{\parallel}$ near the hydrophobic surface increases more than near the hydrophilic surface, because the motion of water molecules is more hindered at the hydrophilic surface[22]. Considering that the stishovite surface is more hydrophilic than graphene, charged ions are less stabilized, so we found more $CO_2$(aq) in the stishovite-confined solutions than in the graphene-confined solutions. This comes in addition to the reactions between $SiO_2$ and solvent molecules, which make the fluids more acidic and thereby lead to destabilization of $HCO_3^-$ in favor of $CO_2$(aq). Therefore, the $CO_2$ concentration increase in the stishovite-confined solutions is a combined result of the hydrophilic confinement and the adsorption preference of $OH^-$ on the stishovite (100) surface.

In our simulations, we used the semilocal Perdew-Burke-Ernzerhof (PBE) exchange-correlation (xc) functional[53], which was reported insufficient to describe aqueous systems at ambient conditions[54]; however, our previous studies showed that PBE performed better for the equation of state and dielectric properties of water[55,56] and the carbon speciation in water[7] at extreme P-T conditions than at ambient conditions. Particularly, we compared the simulations using PBE and a hybrid xc functional, PBE0[57]. For an aqueous carbon solution at ~11 GPa and 1000 K, whose initial mole fraction of $CO_2$(aq) is 0.016, both PBE and PBE0 suggest that $HCO_3^-$ is the dominant carbon species, and its mole percents are 79.8% and 75.0%, respectively[7]. Both PBE and PBE0 lack van der Waals (vdW) interactions, so we performed an additional simulation using the RPBE xc functional[58] with Grimme's D3 vdW corrections and the Becke-Johnson damping (RPBE-D3)[59]. For the solution confined by graphene at 10 GPa and 1000 K, we found that the mole percents of carbon species change by <6% (see Supplementary Fig. 4 and Supplementary Table IV). Particularly, the RPBE-D3 simulation gives that the concentration of $CO_2$(aq) is 0%, while it is $1.3 \pm 0.9\%$ at the PBE level, indicating that our main conclusion that nanoconfinement enhances reactivity of $CO_2$ is not affected by the neglect of the vdW corrections. VdW interactions do not play a major role in breaking and forming of covalent bonds, so do not much affect the chemical speciation studied here.

In summary, we performed extensively long AIMD simulations to study the chemical reactions and speciation of aqueous carbon solutions nanoconfined by graphene and stishovite at 10 GPa and 1000 - 1400 K. We found that the graphene nanoconfinement promotes the $CO_2$(aq) reactions. When graphene is replaced by stishovite, less $CO_2$(aq) reacts, but still more than in the bulk solutions. We found that contacting the stishovite (100) surface makes the solutions more acidic, which shifts the chemical equilibria, though the stishovite surface also catalyzes the $CO_2$(aq) reactions by adsorbing $HCO_3^-$ and $H^+$.

The enhanced reactivity of $CO_2$(aq) in nanoconfinement has important implications for carbon transport and fluid-rock interactions in deep Earth. Aqueous fluids located at grain boundaries in minerals can either exist in isolated fluid-filled pores, or form a connected network of channels along grains facilitating fluid transport[13]. It is known that adding molecular $CO_2$(aq) to water increases the rock-fluid-rock dihedral angle $\theta$, which inhibits fluid flow[60,61]. However, our study shows that $CO_2$(aq) reacts with water under nanoconfinement and also reacts with the solid interface, which may decrease $\theta$ and promote the interconnectivity of fluids[14]. Our study also sheds light on

atomistic mechanisms of $CO_2$ storage through mineral carbonation. $CO_2$ reacts more in nanoconfined water, which benefits $CO_2$ mineralization. If we choose minerals with larger points of zero charge than that of $SiO_2$, such as forsterite[62] and magnesium oxide[63], the $CO_2$ reactivity may be further enhanced.

## Methods

We carried out Born-Oppenheimer ab initio molecular dynamics using the Qbox package[64]. We used periodic boundary conditions and employed plane-wave basis sets and norm-conserving pseudopotentials[65,66], with a plane-wave cutoff of 85 Ry. The cutoff was increased to 145 Ry for pressure calculations. We applied density functional theory and the PBE exchange-correlation functional[53]. We sampled the Brillouin zone at the Γ point. We performed AIMD simulations in the canonical, i.e., NVT, ensemble. Stochastic velocity rescaling was used to control the temperature[67], with a damping factor of 24.2 fs. We replaced hydrogen by deuterium to use a large time step of 0.24 fs in the simulations, but still referred to these atoms as hydrogen atoms.

We ran simulations for 180–480 ps after 20 ps equilibration to reach chemical equilibria (see Supplementary Table I for the simulation details). We analyzed the AIMD trajectories to determine the nature of carbon-containing molecules. For each carbon atom, we searched for the three nearest oxygen atoms, and sorted the C-O distances in increasing order. If the difference between the third and second C-O distance is <0.4 Å, the carbon species is a $CO_3^{2-}$ ion; otherwise, it is $CO_2$. Hydrogen atoms were considered being bonded to their nearest-neighbor oxygen atoms. For the solutions confined by stishovite, the oxygen atoms were considered bonded to silicon atoms when the interatomic distance fell within the first peak of the Si-$O_{aq}$ radial distribution function (RDF), i.e., 2.6 Å, as shown in Supplementary Fig. 5. We also varied the cutoff distances (0.4 and 2.6 Å) by ± 10%, and found that the changes of species concentrations are within the statistical fluctuations of our AIMD simulations (see Supplementary Tables V and VI).

## Data availability

Input files and source data are provided in the repository: https://github.com/nstolte01/confined_aqueous_co2.

## Code availability

Qbox is a free and open source code available at http://qboxcode.org.

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

## Acknowledgements

N.S. acknowledges the Hong Kong Ph.D. Fellowship Scheme. D.P. acknowledges support from the Croucher Foundation through the Croucher Innovation Award, Hong Kong Research Grants Council (Projects GRF-16307618, GRF-16306621, and C6021-19EF), National Natural Science Foundation of China (Project 11774072 and Excellent Young Scientists Fund), the Alfred P. Sloan Foundation through the Deep Carbon Observatory, and the Hetao Shenzhen/Hong Kong Innovation and Technology Cooperation (HZQB-KCZYB-2020083). Part of this work was carried out using computational resources from the National Supercomputer Center in Guangzhou, China.

## Author contributions

D.P. and N.S. designed the research. Calculations were performed by N.S. R.H. implemented the van der Waals dispersion corrections into the Qbox code. N.S. and D.P. contributed to the analysis and discussion of the data and the writing of the manuscript.

## Competing interests

The authors declare no competing interests.
