## [Peer Review File · Nature Communications]

REVIEWER COMMENTS

Reviewer #1 (Remarks to the Author):

The manuscript reports a computational study on the reactivity of carbon dioxide in nanoconfined supercritical water by means of ab initio molecular dynamics. The results reported in the paper can be of great utility in the field of geochemistry and the chemical composition of geological fluids in inner Earth regions: first principle calculations have proven to be a fundamental tool in describing the properties of materials at extreme conditions and the authors contribute in that direction, as previous studies on nanoconfined water at high P/T never focused on geological fluids. Furthermore, to my knowledge, first principles calculations at geologically relevant thermodynamic conditions only investigated bulk water, which can limit the understanding of the phenomena involved. I consider manuscript well prepared and written, and the topic and methodology of this paper to be suitable for Nature Communications, although the authors should address the following points before publications:

1) Why did the authors consider specifically graphene as one of the confining systems for supercritical water? As far as I know, no graphene is found in upper mantle. If the idea was to somehow model graphite layers, the distance between graphene sheets is clearly too large. Instead, if the idea was to model some interstitial space within bulk graphite, the authors should comment on how graphene sheets can substantially differ from graphite layers and that the absence of a bulk phase can modify the behavior of CO₂ solutions. In any case, I recommend that the authors motivate the presence of graphene as case study.

2) Ab Initio MD simulations are notoriously expensive and hardly able to simulate more than 200 atoms, especially for the length involved in the present work. From SI, I see that 5 CO₂ molecules are dissolved in 22 H₂O molecules, which I am afraid can be too few to fully capture the behavior of CO₂ in supercritical water. Have the authors performed any benchmark on the initial molar fraction? They should comment on this.

3) Since bulk phase simulations are used as a term of comparison, it seems that the details of these are missing (even in the SI): how many atoms/molecules? What was the initial molar fraction of CO₂? This is crucial in comparing the results, as the initial conditions can be substantially different.

4) I think that the authors should be more clear in explaining the simulation set up for the stishovite case: from SI it appears that three stishovite layers were used; how many atoms are we talking about? Since this can drastically affect the cost of the simulation (especially when running hundreds

of ps), did the authors use a fixed atoms approach, i.e. fixing a number of atoms corresponding to the bulk of stishovite? Because they mention that the positions of Si atoms in the middle layer were fixed, but it is not clear why fixing only those atoms and not, for example, the whole third layer.

5) Considering eq. 2 and how the acidity of carbon solutions was obtained: the authors mention that CO₂ can react with water to generate H₃O⁺. After this information, which I consider correct, I assume that the concentration of OH⁻ was always zero, or maybe very small for a short part of the simulation when water molecules react with the stishovite surface, as the author described in p. 6 of the manuscript. What concentration of OH⁻ did the author adopt for such equation? The authors should spend a few words (maybe in the SI) in detailing how the $f = \text{pH} - \text{pOH}$ was determined.

5b) Also, as the authors mentioned, since neutrality condition changes at high P-T, what values did the authors use as water self dissociation constant for the conditions involved in the simulations?

6) Although I can agree that plain PBE can be adopted at extreme P-T with a negligible loss in accuracy, as shown in previous works, I am a bit concerned about the lack of comments related to dispersion correction or van der Waals interactions which can be important in modeling water behavior and its hydration properties. Have the authors performed any test on this? I am not familiar with Qbox performances, but empirical correction proposed by Grimme et al. [J. Chem. Phys. 132, 154104 (2010)] can usually be included without a dramatic increase in computational effort. Authors should comment the neglect of vdW correction in their calculations and how does it affect their results.

Minor technical details:

1) The authors mention in the methods section that they ran simulations for 180-480 ps. I think this information is too generic: can they specify which simulation was ran for how long?

2) The radial distribution functions reported in Fig. S3 were obtained for the same number of molecules reported in the main text? If yes, were they smoothed in some way? Because considering the small number of water molecules involved I am surprised on how smooth these are.

3) The identification of different carbon-bearing species, and the bonding of oxygen atoms to stishovite surface, is calculated involving a steplike cutoff distance. Can the author check this assumption? What are the authors thoughts on adopting a softer switching function to monitor the coordination number the atomic species involved? For example, considering the RDF in Fig. S3, it

seems to vary with continuity and never goes to zero between the two peaks (the authors considered a sharp 2.6 Å as a cutoff distance to consider oxygen atoms bonded to silicon ones). I think the authors should comment on this aspect as the persistence and stability of a given carbon species is a central point in the manuscript.

Reviewer #2 (Remarks to the Author):

This manuscript is focused on using first principles molecular dynamics simulations to examine the reactivity of aqueous CO₂ solutions under conditions relevant to the deep earth, including temperature, pressure and the effect of nanoscale confinement by two representative surfaces. The main finding from this investigation is that nanoconfinement with both graphene and stishovite surfaces can cause CO₂ to be more reactive than it would be in the bulk under similar temperatures and pressures. The implication here is that the reactivity of CO₂ in the deep earth may be more complex than previously thought. These findings also indicate that confining CO₂ in nanoporous materials might provide an approach for enhancing the efficiency of mineral carbonization strategies.

Overall, I found this manuscript to be quite interesting and I believe the findings are novel and important. I'd also like to complement the authors on preparing a high quality manuscript that was a pleasure to read. That being said, there are a couple of areas that I think would need further clarification before I can recommend publication of this article.

1. If I understand correctly, the simulations of CO₂ confined with graphene were actually hybrid classical/DFT simulations while the stishovite and bulk simulations were all carried out with standard DFT-based first principles molecular dynamics. If this is true, then I'm puzzled why the rationale for doing this isn't discussed at all in the paper. Is graphene not described well with the PBE functional? Was this an arbitrary choice? Why was a rather different simulation approach used here? Given that the main focus of this paper is to examine confinement effects on the solutions, it's rather strange to have used an approximation like this without fully justifying it and/or demonstrating that it is a reasonable approximation for the graphene surface.

2. It was nice to see that rather long simulation trajectories were used to equilibrate the chemical reactions, but I'm somewhat concerned about the small system sizes. Does it really make sense to refer to the pH of a simulation that only involves 5 CO₂ and 22 H₂O molecules? I'm left wondering if the distributions shown in Figure 2 and Table 2 are strongly influenced by the small system sizes. To be clear, I'm referring to the surface area that was used to confine the liquid, not the confinement

distance. This may be prohibitively expensive to do, but it would be useful to see how sensitive these distributions are to system size effects.

3. The confinement distances examined are very small and different for the two confining surfaces (~9 Angstrom for graphene and ~7 Angstrom for stishovite). Is there any reason why the specific confinement distances were used? These distances are certainly going to be on the extreme end of the pore distributions found in nanoporous minerals. Are the chemical species distributions representative of what you expect for larger confinement distances? A discussion of the differences between surface and nanoconfinement effects might be helpful to include.

Reviewer #3 (Remarks to the Author):

The manuscript entitled "Nanoconfinement Facilitates Reactions of Carbon Dioxide in Supercritical Water" report the promoted reactions of carbon dioxide in critical water under confined environment. They have adopted the theoretical models in two confined environments. The first one is the confined water by graphene sheets. The second model is more realistic which model the water confined by the stishovite. The authors provide clear evidence that the reaction between CO₂ and water is enhanced compared to the bulk solutions. The conclusion is interesting and important for both the environmental science and geology. In the following, I have raised a few questions which could help the authors to further improve the quality of the research and presentation. I would like to recommend its publication after the authors successfully address those issues or questions.

1. On page 5, the author wrote that "Here, the produced OH⁻ ions from the water self-ionization may readily react with CO₂(aq) (reaction (1)), and the enhancement of ϵ also further stabilizes the generated HCO₃⁻ and CO₃²⁻ ions". The above argument will be more convincing if the authors could provide evidence in the supporting material. In the molecular dynamics trajectories, the OH⁻ ions from self-ionization can be extracted which can be further traced in the reaction as stated above.

2. On page 6, the authors wrote that "The hydroxyl groups at the stishovite surface may react with CO₂(aq) to form HCO₃⁻(Fig.5(a))". Again, similar to point 1, it would be more convincing in science if the authors could provide evidence supporting the reaction.

3. In several places, the authors argue that the increased dielectric constant help to stabilize the ions. The authors might want to provide further physical argument to make the above statement more transparent for general readers. It is understood that the Coulombic interaction among ions will be more screened by the larger dielectric constant.

4. I did not find the definition of math symbol of ϵ_{\parallel} . I figured out that it probably means the dielectric constant parallel to the confined surface. I would suggest the author find a place to define it.

5. The readers might be very curious about what is the common features in both confined water that promote the CO₂ reaction. From the paper, I have the impression that the large dielectric constant plays a key role. Could the authors discuss more on this point in the paper?

6. In many places, the authors used the word "may". It gives the audience that the authors are not sure or uncertain about the underlying mechanism. I would suggest the authors to revise those places.

7. The promoted reaction is based on the comparison with bulk solution. Could the authors provide more simulation details on the bulk solutions?

Reviewer #1 (Remarks to the Author):

The manuscript reports a computational study on the reactivity of carbon dioxide in nanoconfined supercritical water by means of ab initio molecular dynamics. The results reported in the paper can be of great utility in the field of geochemistry and the chemical composition of geological fluids in inner Earth regions: first principle calculations have proven to be a fundamental tool in describing the properties of materials at extreme conditions and the authors contribute in that direction, as previous studies on nanoconfined water at high P/T never focused on geological fluids. Furthermore, to my knowledge, first principles calculations at geologically relevant thermodynamic conditions only investigated bulk water, which can limit the understanding of the phenomena involved. I consider manuscript well prepared and written, and the topic and methodology of this paper to be suitable for Nature Communications, although the authors should address the following points before publications:

A: We thank the reviewer for the comments.

1) Why did the authors consider specifically graphene as one of the confining systems for supercritical water? As far as I know, no graphene is found in upper mantle. If the idea was to somehow model graphite layers, the distance between graphene sheets is clearly too large. Instead, if the idea was to model some interstitial space within bulk graphite, the authors should comment on how graphene sheets can substantially differ from graphite layers and that the absence of a bulk phase can modify the behavior of CO₂ solutions. In any case, I recommend that the authors motivate the presence of graphene as case study.

A: Our purpose is not to model graphite. We agree with the reviewer that no graphene is found in the upper mantle so far, so we wrote that stishovite is a “realistic mineral in deep Earth”, but if we only study the stishovite confined solutions, we cannot distinguish the roles of spatial confinement and surface chemistry, both of which affect confined solutions. Because the graphene sheets remain intact without any interface reactions, only the spatial confinement takes effect, which is a good comparison with the stishovite confinement; that is, we compared how the hydrophobic (graphene) and hydrophilic (stishovite) surfaces affect the nanoconfined solutions. In addition, the nanoconfinement by rigid hard walls, such as graphene, hBN, and MoS₂, recently stimulates lots of interest in the chemical physics and physical chemistry communities. Thanks to the rapid development in the fabrication and characterization of 2D materials in recent years, experimentalists are now able to delicately measure the properties of aqueous solutions under graphene nanoconfinement (see, e.g., papers below). Hence, we hope that our simulations can also attract many follow-up experimental studies.

- Algara-Siller, G. *et al.* Square ice in graphene nanocapillaries. *Nature* **519**, 443–445 (2015).
- Chen, J., Schusteritsch, G., Pickard, C. J., Salzmann, C. G. & Michaelides, A. Two dimensional ice from first principles: Structures and phase transitions. *Phys. Rev. Lett.* **116**, 025501 (2016).
- Ruiz-Barragan, S., Muñoz-Santiburcio, D., Körning, S. & Marx, D. Quantifying anisotropic dielectric response properties of nanoconfined water within graphene slit pores. *Phys. Chem. Chem. Phys.* **22**, 10833–10837 (2020).
- Fumagalli, L. *et al.* Anomalously low dielectric constant of confined water. *Science* **360**, 1339–1342 (2018).

On Page 4, we added our motivation: “Although graphene is not found in deep Earth so far, it provides a good comparison with stishovite. In graphene confinement, there are no chemical reactions between graphene and solutions, whereas the dangling atoms in stishovite actively

participate in aqueous carbon reactions, so we can compare the effects of spatial confinement with and without interface chemistry. What's more, thanks to the rapid development in the fabrication and characterization of 2D materials in recent years, experimentalists are now able to delicately measure the properties of aqueous solutions under graphene nanoconfinement, so we hope our study can also attract many follow-up experiments.”

2) Ab Initio MD simulations are notoriously expensive and hardly able to simulate more than 200 atoms, especially for the length involved in the present work. From SI, I see that 5 CO₂ molecules are dissolved in 22 H₂O molecules, which I am afraid can be too few to fully capture the behavior of CO₂ in supercritical water. Have the authors performed any benchmark on the initial molar fraction? They should comment on this.

A: We thank the reviewer for understanding the computational expense of AIMD simulations. We doubled the molecule numbers in the simulation box (i.e., 10 CO₂ molecules dissolved in 44 H₂O molecules), and adjusted the x- and y-dimensions of the unit cell accordingly while keeping the confinement width unchanged, and conducted an additional 50 ps simulation for the graphene confined solution at ~10 GPa and 1000 K. The 50 ps simulation should be long enough, because the 200 ps simulation in Supplementary Fig. 3 shows that the concentrations of reaction products do not change considerably after 50 ps (see Fig. 1 below). We found that after increasing the size of the simulation box, the mole percents of carbon species change by less than 2.5%, indicating that the original simulation size is big enough. The carbon reactions in the confinement at extreme P-T conditions studied here typically involve three to four molecules in the xy plane, which can be well captured by our simulations with periodic boundary conditions. The x and y dimensions of our simulation box are 1.1 nm and 0.9 nm, respectively, which are similar to the cubic box dimension in the bulk simulation, 1.1 nm. The small number of molecules in our simulations is mainly due to the confinement size along the z direction (< 1 nm). In addition, the molecular correlation length at the high temperatures is shorter than at ambient conditions, so the convergence size is also smaller.

In our simulations, the initial mole fraction was kept at 0.185, so we can directly compare the solutions with the same carbon concentration under different confinements. In the stishovite confinement simulations, we have 5 CO₂ molecules dissolved in 22 H₂O molecules, and also 24 SiO₂ molecules, while in the graphene confinement simulations, we replaced SiO₂ by the graphene model potential to have a fair comparison.

We added Supplementary Fig. 4 and Supplementary Table IV, which compare the simulations with small and big unit cells, to the supplementary information.

Figure 1: The 1 ps window average of the mole percents of aqueous carbon species in the graphene confinement at ~10 GPa and 1000 K. The initial mole fraction of CO₂(aq) is 0.185. (a) There are 5 CO₂ and 22 H₂O molecules in one unit cell. The exchange-correlation functional

is PBE. (b) There are 10 CO₂ and 44 H₂O molecules in one unit cell. The exchange-correlation functional is PBE. (c) There are 5 CO₂ and 22 H₂O molecules in one unit cell. The exchange-correlation functional is RPBE with the D3 van der Waals correction.

3) Since bulk phase simulations are used as a term of comparison, it seems that the details of these are missing (even in the SI): how many atoms/molecules? What was the initial molar fraction of CO₂? This is crucial in comparing the results, as the initial conditions can be substantially different.

A: We added the details of bulk phase simulations in Supplementary Table I. When the temperature is 1000 K, the pressure is ~10 GPa, and the initial mole fraction of CO₂ is 0.185, so the initial conditions are the same as those under confinement. At 1400 K, we did the bulk simulations with the initial mole fractions of 0.032, 0.333, and 0.600, and then interpolated the results using cubic splines to obtain the mole fraction of 0.185. We always compared the solutions with the same initial mole fraction and the same P-T conditions.

4) I think that the authors should be more clear in explaining the simulation set up for the stishovite case: from SI it appears that three stishovite layers were used; how many atoms are we talking about? Since this can drastically affect the cost of the simulation (especially when running hundreds of ps), did the authors use a fixed atoms approach, i.e. fixing a number of atoms corresponding to the bulk of stishovite? Because they mention that the positions of Si atoms in the middle layer were fixed, but it is not clear why fixing only those atoms and not, for example, the whole third layer.

A: Each stishovite layer has 8 Si atoms and 16 O atoms, so there are 24 Si atoms and 48 O atoms with periodic boundary conditions to model stishovite layers in the simulation box. The reviewer's concern about the simulation cost is very valid, so we did not include too many solution molecules to run for hundreds of ps.

We fixed the Si atoms in the middle layer of the SiO₂ slab. It is not a good idea to fix the whole third layer, because it will break the thermal equilibrium. Given that the third layer is in touch with solutions, if we fix the whole third layer, which means the temperature of the third layer is 0 K, heat will flow from the solutions at 1000 or 1400 K to the stishovite layer, and it may affect the chemical speciation in the solutions.

On Page S4 in the supplementary information, we revised the simulation setup for the stishovite: "For stishovite-confined solutions, the stishovite slab is made by three stoichiometric layers of SiO₂ exposing the low-energy (100) surface to solutions (see Supplementary Fig. 2). Each SiO₂ layer contains 8 SiO₂ formula units, so there are 24 silicon atoms and 48 oxygen atoms in the unit cell. In the NVT simulations, we only fixed the positions of silicon atoms in the middle layer of the stishovite slab. Table I summarizes the simulation setups.". We added Supplementary Fig. 2 to show the structure of the stishovite slab.

5) Considering eq. 2 and how the acidity of carbon solutions was obtained: the authors mention that CO₂ can react with water to generate H₃O⁺. After this information, which I consider correct, I assume that the concentration of OH⁻ was always zero, or maybe very small for a short part of the simulation when water molecules react with the stishovite surface, as the author described in p. 6 of the manuscript. What concentration of OH⁻ did the author adopt for such

equation? The authors should spend a few words (maybe in the SI) in detailing how the $f = \text{pH} - \text{pOH}$ was determined.

A: Indeed, the concentration of OH^- in our simulations is small, but not zero, due to the dissociation of water at the stishovite surface as well as the self-ionization in the solutions. To obtain $\text{pH} - \text{pOH}$, we only need the concentration ratio: $[\text{H}_3\text{O}^+]/[\text{OH}^-] = N_{\text{H}_3\text{O}^+}/N_{\text{OH}^-}$, where $N_{\text{H}_3\text{O}^+}$ and N_{OH^-} are the average numbers of the H_3O^+ and OH^- ions, respectively, in one unit cell per each AIMD snapshot. For example, $N_{\text{OH}^-} = \frac{1}{m} \sum_{i=1}^m n_{\text{OH}^-}(i)$, where $n_{\text{OH}^-}(i)$ is the number of the OH^- ions in the i th AIMD snapshot, and m is the total number of AIMD snapshots.

On Page S4 in the supplementary information, we explained how f was determined.

5b) Also, as the authors mentioned, since neutrality condition changes at high P-T, what values did the authors use as water self dissociation constant for the conditions involved in the simulations?

A: We did not use the water self-dissociation constant, because we only need the $[\text{H}_3\text{O}^+]/[\text{OH}^-]$ ratio. If $f=0$, the solution is neutral.

6) Although I can agree that plain PBE can be adopted at extreme P-T with a negligible loss in accuracy, as shown in previous works, I am a bit concerned about the lack of comments related to dispersion correction or van der Waals interactions which can be important in modelling water behavior and its hydration properties. Have the authors performed any test on this? I am not familiar with Qbox performances, but empirical correction proposed by Grimme et al. [J. Chem. Phys. 132, 154104 (2010)] can usually be included without a dramatic increase in computational effort. Authors should comment the neglect of vdW correction in their calculations and how does it affect their results.

A: We implemented Grimme's D3 corrections into the Qbox code, as suggested by the reviewer, and performed an additional 50 ps simulation using the RPBE+D3 functional for the solution confined by graphene at 10 GPa and 1000 K (Supplementary Fig. 4 and Supplementary Table IV). We found that the mole percents of carbon species change by less than 6%. Particularly, the RPBE+D3 simulation gives that the concentration of $\text{CO}_2(\text{aq})$ is $\sim 0\%$, while it is $1.3 \pm 0.9\%$ at the PBE level, indicating that our main conclusion that the nanoconfinement enhances the reactivity of CO_2 is not affected by the neglect of the vdW corrections. VdW interactions do not play a major role in breaking and forming of covalent bonds, so do not much affect the chemical speciation studied here.

We have included a discussion of the effect of van der Waals interactions in the main text (Page 10), and include the results of the RPBE+D3 simulations in Supplementary Fig. 4 and Supplementary Table IV.

Minor technical details:

1) The authors mention in the methods section that they ran simulations for 180-480 ps. I think this information is too generic: can they specify which simulation was ran for how long?

A: We included the length of each simulation in Supplementary Table I.

2) The radial distribution functions reported in Fig. S3 were obtained for the same number of molecules reported in the main text? If yes, were they smoothed in some way? Because considering the small number of water molecules involved I am surprised on how smooth these are.

A: Yes, they were obtained using the same number of molecules reported in the main text (Supplementary Table I). We did not smooth them. Because we have a great number of AIMD snapshots, the radial distribution functions look very smooth. Our AIMD time step is 10 a.u = 0.24 fs. The simulation at 1000 K is ~480 ps long, containing about 2.0×10^6 MD snapshots, and the 280 ps simulation at 1400 K has about 1.2×10^6 MD snapshots. We used the whole trajectories to compute the radial distribution functions.

3) The identification of different carbon-bearing species, and the bonding of oxygen atoms to stishovite surface, is calculated involving a steplike cutoff distance. Can the author check this assumption? What are the authors thoughts on adopting a softer switching function to monitor the coordination number the atomic species involved? For example, considering the RDF in Fig. S3, it seems to vary with continuity and never goes to zero between the two peaks (the authors considered a sharp 2.6 Å as a cutoff distance to consider oxygen atoms bonded to silicon ones). I think the authors should comment on this aspect as the persistence and stability of a given carbon species is a central point in the manuscript.

A: We tested whether our results are sensitive to the choice of cutoff distances by varying the cutoffs by $\pm 10\%$ and re-analyzing the speciation of carbon-containing compounds for $\text{CO}_2(\text{aq})$ confined in stishovite at 10 GPa and 1000 K. For all carbon-containing molecules, the differences between the new mole percents and the original ones are within the statistical fluctuations of simulations, indicating that our results are not affected by the steplike cutoff (see Supplementary Table V and Supplementary Table VI).

We used the difference (δ) between the third and second nearest C-O distances to distinguish CO_2 and CO_3^{2-} . The figure below shows the distribution of δ in the bulk solution of $\text{CO}_2(\text{aq})$ at ~10 GPa and 1000 K, whose initial mole fraction of $\text{CO}_2(\text{aq})$ is 0.185. The probability density between 0.4 and 0.9 is very close to 0, so varying δ would not affect the mole percents of carbon species (see Supplementary Table VI).

More importantly, as long as we stick to a consistent definition of molecular geometry, the comparison between different solutions is valid.

In the Method section, we added “we also varied the cutoff distances (0.4 Å and 2.6 Å) by $\pm 10\%$, and found that the changes of species concentrations are within the statistical fluctuations of AIMD simulations (see Supplementary Tables V and VI).”

Figure 2. The probability density of $\delta = d_3 - d_2$, where d_2 is the distance between a central C atom and its second-nearest O atom, and d_3 is the distance from the C atom to its third-nearest O atom. If $\delta \leq 0.4 \text{ \AA}$, the three nearest O atoms are bonded to the central C atom; otherwise only the two nearest O atoms are bonded.

Reviewer #2 (Remarks to the Author):

This manuscript is focused on using first principles molecular dynamics simulations to examine the reactivity of aqueous CO₂ solutions under conditions relevant to the deep earth, including temperature, pressure and the effect of nanoscale confinement by two representative surfaces. The main finding from this investigation is that nanoconfinement with both graphene and stishovite surfaces can cause CO₂ to be more reactive than it would be in the bulk under similar temperatures and pressures. The implication here is that the reactivity of CO₂ in the deep earth may be more complex than previously thought. These findings also indicate that confining CO₂ in nanoporous materials might provide an approach for enhancing the efficiency of mineral carbonization strategies.

Overall, I found this manuscript to be quite interesting and I believe the findings are novel and important. I'd also like to complement the authors on preparing a high quality manuscript that was a pleasure to read. That being said, there are a couple of areas that I think would need further clarification before I can recommend publication of this article.

A: We thank the reviewer for the comments.

1. If I understand correctly, the simulations of CO₂ confined with graphene were actually hybrid classical/DFT simulations while the stishovite and bulk simulations were all carried out with standard DFT-based first principles molecular dynamics. If this is true, then I'm puzzled why the rationale for doing this isn't discussed at all in the paper. Is graphene not described well with the PBE functional?

A: No, it is known that the PBE functional cannot describe graphene well, due to the lack of the van der Waals interaction (see, e.g., the paper below).

- Ma, J. et al. Adsorption and diffusion of water on graphene from first principles. *Phys. Rev. B* 84, 033402 (2011)

Was this an arbitrary choice? Why was a rather different simulation approach used here? Given that the main focus of this paper is to examine confinement effects on the solutions, it's rather strange to have used an approximation like this without fully justifying it and/or demonstrating that it is a reasonable approximation for the graphene surface.

A: It is not an arbitrary choice. To obtain the correct van der Waals interactions between graphene and solutions, we used the interaction potential that is fitted to the data calculated by diffusion quantum Monte Carlo and van der Waals density functional theory. Quantum Monte Carlo is one of the most accurate methods to calculate the van der Waals interaction (PRB 84, 033402 (2011) and Ref. 42 in the main text), and is also used to develop approximated exchange-correlation functionals in density functional theory.

In addition, there is no surface chemistry or charge transfer between the model potential and solutions, so by comparing the model potential with the more realistic stishovite slab, we can distinguish the effects of spatial confinement and surface chemistry.

Using model potentials to create nanoconfinement for aqueous solutions has been adopted in many previous studies (see, e.g., papers below).

- Koga, K, Tanaka, H. & Zeng, X. C. First-order transition in confined water between high-density and low-density amorphous phases. *Nature* **408**, 564–567 (2000).
- Chen, J., Schusteritsch, G., Pickard, C. J., Salzmann, C. G. & Michaelides, A. Two dimensional ice from first principles: Structures and phase transitions. *Phys. Rev. Lett.* **116**, 025501 (2016).
- Munoz-Santiburcio, D. & Marx, D. Nanoconfinement in slit pores enhances water self-dissociation. *Phys. Rev. Lett.* **119**, 056002 (2017).

We added the justification on Page S2 in the supplementary information.

2. It was nice to see that rather long simulation trajectories were used to equilibrate the chemical reactions, but I'm somewhat concerned about the small system sizes. Does it really make sense to refer to the pH of a simulation that only involves 5 CO₂ and 22 H₂O molecules?

A: It is not very good to refer to the pH or pOH, so we compared the difference between pH and pOH, ie., $f = \text{pH} - \text{pOH} = [\text{H}_3\text{O}^+]/[\text{OH}^-] = N_{\text{H}_3\text{O}^+}/N_{\text{OH}^-}$, where $N_{\text{H}_3\text{O}^+}$ and N_{OH^-} are the average numbers of the H₃O⁺ and OH⁻ ions, respectively, in one unit cell per each AIMD snapshot. It is true that we have only 5 CO₂ and 22 H₂O molecules in one unit cell in the confinement simulations, but we have more than 10⁶ MD snapshots. The time average of the concentration ratio [H₃O⁺]/[OH⁻] is well converged after long simulations. We also tested the size dependence (see below).

We added Supplementary Table VII to show calculated $N_{\text{H}_3\text{O}^+}$ and N_{OH^-} .

I'm left wondering if the distributions shown in Figure 2 and Table 2 are strongly influenced by the small system sizes. To be clear, I'm referring to the surface area that was used to confine the liquid, not the confinement distance. This may be prohibitively expensive to do, but it would be useful to see how sensitive these distributions are to system size effects.

A: We thank the reviewer for understanding the computational expense of AIMD simulations. We doubled the molecule numbers in the simulation box (10 CO₂ molecules dissolved in 44 H₂O molecules), while keeping the confinement width unchanged, and conducted an additional 50 ps simulation for the graphene confined solution at ~10 GPa and 1000 K. The 50 ps simulation should be long enough, because the 200 ps simulation in Supplementary Fig. 3 shows that the concentrations of reaction products do not change considerably after 50 ps. We found that after increasing the size of the simulation box, the mole percents of carbon species change by less than 2.5%, indicating that the original simulation size is big enough (see the figure below).

Figure 3: The 1 ps window average of the mole percents of aqueous carbon species in the graphene confinement at ~10 GPa and 1000 K. The initial mole fraction of CO₂ is 0.185. (a)

There are 5 CO₂ and 22 H₂O molecules in one unit cell. The exchange-correlation functional is PBE. (b) There are 10 CO₂ and 44 H₂O molecules in one unit cell. The exchange-correlation functional is PBE. (c) There are 5 CO₂ and 22 H₂O molecules in one unit cell. The exchange-correlation functional is RPBE with the D3 van der Waals correction.

On Page S3 in the supplementary information, we added “the simulation details are shown in Supplementary Table I. At ~10 GPa and 1000 K, we also doubled the number of molecules in the unit cell and added the van der Waals corrections to test our computational setups. We used the SG15 Optimized Norm-Conserving Vanderbilt (ONCV) pseudopotentials with a plane-wave cutoff of 65 Ry to speed up calculations. Supplementary Fig. 4 and Supplementary Table IV show that our simulation sizes in Supplementary Table I are big enough, and the van der Waals corrections change the chemical speciation little.”

3. The confinement distances examined are very small and different for the two confining surfaces (~9 Angstrom for graphene and ~7 Angstrom for stishovite). Is there any reason why the specific confinement distances were used? These distances are certainly going to be on the extreme end of the pore distributions found in nanoporous minerals.

A: The specific confinement distances were obtained by the simulations at the constant pressure of ~10 GPa and 1000 K; the P-T condition is typically found at the bottom of Earth’s upper mantle. The graphene model is hydrophobic, while the stishovite slab is hydrophilic. Because there is a thin vacuum between the hydrophobic surface and water, the confinement width with graphene is larger than that with stishovite. The previous studies suggest that the confinement width between grain boundaries in silicates is in the range of 4~12 Angstrom (reference below), so our choice is reasonable.

- Marquardt, K. & Faul, U. H. The structure and composition of olivine grain boundaries: 40 years of studies, status and current developments. *Phys. Chem. Miner.* **45**, 139–172 (2018).

Are the chemical species distributions representative of what you expect for larger confinement distances? A discussion of the differences between surface and nanoconfinement effects might be helpful to include.

A: We would expect that at larger confinement distances, the chemical speciation is closer to that in bulk solutions. On Page 6, we added that “When the interlayer distance between graphene sheets increases beyond ~1.5 nm, the bulk behaviour of water is recovered in the center of the slit pore and the effects of nanoconfinement become less obvious.”

We discussed the differences between surface and nanoconfinement on Page 4 and 9. Because we use the model potential to simulate graphene, there are only nanoconfinement effects in the graphene confinement, whereas the stishovite confinement has both surface and nanoconfinement effects.

Reviewer #3 (Remarks to the Author):

The manuscript entitled "Nanoconfinement Facilitates Reactions of Carbon Dioxide in Supercritical Water" report the promoted reactions of carbon dioxide in critical water under confined environment. They have adopted the theoretical models in two confined environments. The first one is the confined water by graphene sheets. The second model is more realistic which model the water confined by the stishovite. The authors provide clear evidence that the reaction between CO₂ and water is enhanced compared to the bulk solutions. The conclusion is interesting and important for both the environmental science and geology. In the following, I have raised a few questions which could help the authors to further improve the quality of the research and presentation. I would like to recommend its publication after the authors successfully address those issues or questions.

A: We thank the reviewer for the comments.

1. On page 5, the author wrote that "Here, the produced OH⁻ ions from the water self-ionization may readily react with CO₂(aq) (reaction (1)), and the enhancement of ϵ also further stabilizes the generated HCO₃⁻ and CO₃⁻² ions". The above argument will be more convincing if the authors could provide evidence in the supporting material. In the molecular dynamics trajectories, the OH⁻ ions from self-ionization can be extracted which can be further traced in the reaction as stated above.

A: We added the reaction snapshots in Supplementary Fig. 7, where CO₂(aq) reacts with OH⁻ to form HCO₃⁻.

2. On page 6, the authors wrote that "The hydroxyl groups at the stishovite surface may react with CO₂(aq) to form HCO₃⁻(Fig.5(a))". Again, similar to point 1, it would be more convincing in science if the authors could provide evidence supporting the reaction.

A: Fig. 5(a) in the manuscript shows the evidence for this reaction. At t=0.0 ps, a hydroxyl group was absorbed on the stishovite surface, at t=0.02 ps, CO₂(aq) started to react with the hydroxyl group, and at t= 0.31 ps, the HCO₃⁻ was formed.

At the bottom of Page 7, we rephrased the sentences describing this finding to be more clear: "In our simulations, we found reactions between hydroxyl groups at the SiO₂ surface and CO₂(aq) in the solutions forming HCO₃⁻. Fig. 5(a) shows the reaction snapshots at 1000 K."

3. In several places, the authors argue that the increased dielectric constant help to stabilize the ions. The authors might want to provide further physical argument to make the above statement more transparent for general readers. It is understood that the Coulombic interaction among ions will be more screened by the larger dielectric constant.

A: We included an explanation of the effect of the dielectric constant on Coulomb interactions on Page 6: "In water, the Coulomb interaction between two ions is $F = \frac{q_1 q_2}{\epsilon_0 r}$, where q_1 and q_2 are the charges of the two ions, and r is their distance. With increasing the dielectric constant, the magnitude of F decreases, so it is easier to separate a cation from an anion."

4. I did not find the definition of math symbol of ϵ_{\parallel} . I figured out that it probably means the dielectric constant parallel to the confined surface. I would suggest the author find a place to define it.

A: It is defined on Page 6: “It has been reported that the dielectric constant of nanoconfined water in the direction parallel to the confining surfaces (ϵ_{\parallel}), increases significantly compared to the bulk value (ϵ_0).”

5. The readers might be very curious about what is the common features in both confined water that promote the CO₂ reaction. From the paper, I have the impression that the large dielectric constant plays a key role. Could the authors discuss more on this point in the paper?

A: Yes, the enhanced dielectric constant indeed plays an important role. On Page 6, we wrote that “It has been reported that the dielectric constant of nanoconfined water in the direction parallel to the confining surfaces (ϵ_{\parallel}) increases significantly compared to the bulk value (ϵ_0)”, and “The enhancement of ϵ_{\parallel} also further stabilizes HCO₃⁻ and CO₃²⁻ ions generated in the reaction between CO₂ and H₂O or OH⁻.” On Page 9, we compared the difference of dielectric constants between graphene- and stishovite-confined solutions: “The nanoconfinement enhances ϵ_{\parallel} , which stabilized charged ions, so in both graphene- and stishovite-confined solutions, more CO₂(aq) reacts than in bulk solutions. However, it has been reported that ϵ_{\parallel} near the hydrophobic surface increases more than near the hydrophilic surface, because the motion of water molecules are more hindered at the hydrophilic surface. Considering that the stishovite surface is more hydrophilic than graphene, charged ions are less stabilized, so we found more CO₂(aq) in the stishovite-confined solutions than in the graphene-confined solutions.”

6. In many places, the authors used the word "may". It gives the audience that the authors are not sure or uncertain about the underlying mechanism. I would suggest the authors to revise those places.

A: We revised those places accordingly.

7. The promoted reaction is based on the comparison with bulk solution. Could the authors provide more simulation details on the bulk solutions?

A: We included the simulation details of bulk solutions in Table SI in the supplementary information.

REVIEWERS' COMMENTS

Reviewer #1 (Remarks to the Author):

The authors have addressed all my comments by performing a suitable amount of additional work to answer all the questions. In my opinion, this version of the manuscript is ready to be published.

Reviewer #2 (Remarks to the Author):

I have read through the authors response to my original concerns and I am happy with the modified version of the paper. I would like to recommend publication of this paper in it's current form.

Reviewer #3 (Remarks to the Author):

Dear Editor,

In this resubmission, I am glad to find that the authors have addressed all the referees' questions very seriously. In particular, they have performed extensive calculations and provided evidences to support several physical points that I requested. Therefore, I recommend its publication.